

# Ferroptosis-related gene transferrin receptor protein 1 expression correlates with the prognosis and tumor immune microenvironment in cervical cancer

Xiujuan Shang[1,*], Hongdong Wang[2,*], Jin Gu[1], Xiaohui Zhao[1], Jing Zhang[3], Bohao Sun[3] and Xinming Zhu[1]

[1] Department of Laboratory Medicine, Lianyungang Affiliated Hospital of Nanjing University of Chinese Medicine, Lianyungang, Jiangsu, China
[2] Lianyungang Maternal and Child Health Hospital, Lianyungang, Jiangsu, China
[3] Department of Pathology, Second Affiliated Hospital, School of Medicine, Zhejiang University, Hangzhou, Zhejiang, China
* These authors contributed equally to this work.

Corresponding authors
Bohao Sun, 1551031172@qq.com
Xinming Zhu, lygzhuxinming@163.com

## ABSTRACT

**Background:** Ferroptosis is a non-apoptotic iron-dependent form of cell death implicated in various cancer pathologies. However, its precise role in tumor growth and progression of cervical cancer (CC) remains unclear. Transferrin receptor protein 1 (TFRC), a key molecule associated with ferroptosis, has been identified as influencing a broad range of pathological processes in different cancers. However, the prognostic significance of TFRC in CC remains unclear. The present study utilized bioinformatics to explore the significance of the ferroptosis-related gene TFRC in the progression and prognosis of CC.

**Methods:** We obtained RNA sequencing data and corresponding clinical information on patients with CC from The Cancer Genome Atlas (TCGA), Genotype Tissue Expression (GTEx) and Gene Expression Omnibus (GEO) databases. Using least absolute shrinkage and selection operator (LASSO) Cox regression, we then generated a multigene signature of five ferroptosis-related genes (FRGs) for the prognostic prediction of CC. We investigated the relationship between TFRC gene expression and immune cell infiltration by employing single-sample GSEA (ssGSEA) analysis. The potential functional role of the TFRC gene was evaluated through gene set enrichment analysis (GSEA). Immunohistochemistry and qPCR was employed to assess TFRC mRNA and protein expression in 33 cases of cervical cancer. Furthermore, the relationship between TFRC mRNA expression and overall survival (OS) was investigated in patients.

**Results:** CC samples had significantly higher TFRC gene expression levels than normal tissue samples. Higher TFRC gene expression levels were strongly associated with higher cancer T stages and OS events. The findings of multivariate analyses illustrated that the OS in CC patients with high TFRC expression is shorter than in patients with low TFRC expression. Significant increases were observed in the levels of TFRC mRNA and protein expression in patients diagnosed with CC.

**Conclusion:** Increased TFRC expression in CC was associated with disease progression, an unfavorable prognosis, and dysregulated immune cell infiltration. In addition, it highlights ferroptosis as a promising therapeutic target for CC.

## INTRODUCTION

Cervical cancer (CC) is the fourth most prevalent cancer among females, with approximately 600,000 new cases and 300,000 deaths being reported annually worldwide (*Li et al., 2023*; *Wang et al., 2023b*). Over the past few decades, the global incidence and mortality rates of CC have been declining, owing to advancements in early screening, diagnosis, treatment, and vaccination (*Ma et al., 2019*; *Yu et al., 2022*). Regular gynecological screenings, which involve cervical cytology and HPV testing, are pivotal for averting cervical cancer. Nonetheless, these approaches have been ineffective in inducing tumor regression. Persistent infection with HPV significantly increases the risk of developing high-grade cervical intraepithelial neoplasia (*Ma et al., 2022*). However, despite the availability of regular cervical screening and human papillomavirus vaccination as preventive measures, CC remains a substantial global public health challenge (*Bokulich et al., 2022*; *Schüz & Espina, 2021*). Early and accurate identification of patients at high risk of CC recurrence and timely adjustment of treatment strategies, including immunotherapy or targeted therapy, are crucial steps for improving patient prognosis (*Qi et al., 2021*). There is also an urgent need for developing reliable prognostic biomarkers and novel therapeutic strategies. Accumulating evidence suggests that ferroptosis plays a significant role in the pathogenesis and development of resistance to treatment in various malignancies, including CC (*Gianì et al., 2020*; *Wang, Chan & Cho, 2019*). Iron and lipid metabolisms, together with various signaling pathways, regulate ferroptosis, which plays a crucial role in numerous pathophysiological processes (*Yu et al., 2021*; *Zhu & Li, 2023*). Comprehensive treatment options for CC include chemotherapy, radiotherapy, molecular targeted therapy, immunotherapy, traditional Chinese medicine, and nanotechnology-based therapy (*Zhang et al., 2022*; *Zheng et al., 2023*). Early-stage cervical cancer patients generally have a better prognosis when treated with a combination of surgery and radiotherapy or chemotherapy. However, options for advanced-stage cervical cancer treatment are relatively scarce (*Small et al., 2017*). Despite the potential for prolonged patient survival through the use of combined immunotherapy and anti-angiogenesis therapy alongside radiotherapy or chemotherapy, challenges such as platinum resistance, recurrence, and metastasis remain unresolved and necessitate further investigation (*Meijer & Snijders, 2014*). Targeting ferroptosis significantly enhances antitumor immunity and holds promise for treating drug-resistant tumors (*Cai et al., 2023*; *Meng et al., 2023*). For instance, *Zhang et al. (2023)* discovered that eicosapentaenoic acid increases the sensitivity of osteosarcoma cells to cisplatin by inducing ferroptosis, reducing programmed death ligand one expression, and attenuating immune evasion. These findings suggest the potential application of combining immune checkpoint inhibitors in order to target ferroptosis. Additionally, inhibiting ferroptosis in liver cancer cells promotes tumor growth and metastasis, whereas inducing ferroptosis is advantageous to inhibit tumor growth and proliferation (*Lee et al., 2023*; *Zhu & Li, 2023*). FRGs may serve as a promising therapeutic target for CC patients. For example, the ferroptosis-related gene

EPAS1 enhances the proliferation, invasion, and migration of HeLa and SiHa cells, promoting malignant behavior in CC cells (*Lu et al., 2024*). *Tang et al. (2024)* demonstrated that the ferroptosis-related gene TFRC inhibits the proliferation and invasion abilities of T24 and UMUC-3 cells. Additionally, TFRC emerges as a potential novel predictive model for OS and immunotherapy efficacy in bladder cancer patients. However, TFRC's role in CC remains unexplored. Consequently, ferroptosis-based therapy offers a novel approach to enhance the therapeutic effects of cancer chemotherapy and shows promising clinical prospects (*Lei et al., 2021*; *Lin et al., 2022*).

This study aimed to assess the potential of ferroptosis-related genes (FRGs) for personalized prognostic predictions. Utilizing the Genotype Tissue Expression (GTEx) and The Cancer Genome Atlas (TCGA) databases, the clinical significance of TFRC in CC was investigated. Furthermore, the significance was confirmed in a clinical cohort of CC. These findings enhance our understanding of TFRC's role in CC, facilitating protein detection, assessing its clinical importance and prognostic value, and guiding the development of innovative therapeutic approaches for CC patients.

# MATERIALS AND METHODS

## TCGA, GTEx, and GEO datasets

Gene expression quantification data and clinical information on females with CC were downloaded from the TCGA database (*Zhou et al., 2023*). Gene expression data from healthy cervical tissues were obtained from the GTEx database (*Chen et al., 2022*). The validation cohorts, consisting of complete expression profile data (GSE63514 and GSE63678), were obtained from the GEO database (https://www.ncbi.nlm.nih.gov/gds).

## Identification and analysis of differentially expressed genes

Prior to conducting the differential expression analysis, we conducted background correction and quantile normalization utilizing the robust multi-array analysis (RMA) method implemented in the limma R package, resulting in the generation of the normalized gene expression matrix. $P$-value $< 0.05$ and $\log_2$ fold-change $> |1|$ were set as the cut-off criteria to identify the statistically significant differentially expressed genes (DEGs) (*Shen et al., 2020*; *Yi et al., 2021*; *Zhen et al., 2020*).

## Identification of ferroptosis modules using weighted gene co-expression network analysis

We performed gene co-expression network analysis using the R package weighted gene co-expression network analysis (WGCNA) (*Bhatia et al., 2021*; *Shi et al., 2023*). Briefly, we converted the expression levels of individual transcripts into a similarity matrix based on Pearson's correlation values between paired genes. To detect the outliers, we constructed a hierarchical clustering tree based on the expression matrix. To construct a scale-free network, we determined an appropriate soft thresholding power ($\beta$) as 6 by plotting $R^2$ (scale-free topology fitting index) against the soft thresholds. Next, we calculated the intramodular connectivity between genes exhibiting similar expression using the topological overlap dissimilarity measure. Finally, we utilized the dynamic hybrid cutting

method to establish a hierarchical clustering tree and identify the co-expressed gene modules. Ferroptosis scores were obtained by applying Gene Set Variation Analysis to the ferroptosis genes present in the samples.

## Identification of ferroptosis-related prognostic genes using least absolute shrinkage and selection operator Cox regression

In the next part of the study, we used the LASSO logistic regression method to identify the prognostic genes associated with ferroptosis in CC. The 'glmnet' R package was employed for the LASSO feature selection (*Zhang, Pei & Zhu, 2023*). Initially, 10-fold cross-validation was performed to determine the optimal value of the tuning parameter lambda, which influenced the shrinkage penalty applied to the regression coefficients. The risk score signature was defined as the combination of Coefi (representing the coefficient obtained from LASSO Cox regression) and Expri (*Gao et al., 2022*; *Wang et al., 2021*).

## Functional annotation of FRGs

To investigate potential biological processes and signaling pathways associated with the identified FRGs, we conducted Gene Ontology (GO) and Kyoto Encyclopedia of Genes and Genomes (KEGG) enrichment analyses using the R package 'clusterProfiler' (*Ding et al., 2022*). The crucial intersecting genes were subjected to this analysis. GO enrichment analysis elucidated three aspects: biological processes, cellular components, and molecular functions.

## Multi-omics analyses of the FRGs

Mutation data for patients with CC were downloaded from the TCGA database. The relationship between the risk score and gene expression was visualized using the Kaplan–Meier (KM) curve, which was implemented using the KM R package 'survival' (*Zengin & Önal-Süzek, 2021*). Immunohistochemical images obtained from The Human Protein Atlas were used to determine the protein expression levels of the five key ferroptosis-related prognostic genes in both CC and normal cervical tissues (*Xu et al., 2020b*). Immunohistochemistry staining outcomes for five key genes were derived from the Human Protein Atlas (HPA) database.

## Construction and validation of an FRGs-based prognostic signature for CC cohorts

We employed the KM method to evaluate the disparity in survival outcomes between the high-risk and low-risk groups. From the FRGs, we carefully selected those essential for constructing a prognostic risk score model to predict the overall survival (OS) of patients with CC. To confirm the predictive capability of the FRGs-based signature independently, we conducted univariate and multivariate Cox regression analyses using the "survival" package in R. These analyses involved assessing risk scores along with other clinical factors such as T stage, N stage, M stage, and age.

## TFRC expression analysis of prognosis, model development, and assessment

The present study examined prognostic parameters, namely Disease Specific Survival (DSS), and Progress Free Interval (PFI) by analyzing patient data obtained from TCGA. The analysis was conducted within the clinical meaning module of the Xiantao platform (https://www.xiantao.love/), employing Cox regression and Kaplan-Meier methods. The threshold value for categorizing TFRC gene expression into low and high groups was determined based on the median value. To establish the association between clinical-pathological characteristics and TFRC gene expression, we employed the Wilcoxon signed-rank sum test in conjunction with logistic regression. The findings derived from the Cox regression model were then combined with the independent prognostic variables obtained from the univariate analysis. Subsequently, survival probabilities for 1, 3, and 5 years were projected using these integrated data. The accuracy of these projections was evaluated by comparing them to the actual occurrences through calibration curves.

## Analysis of the infiltration of immune cells

We applied the ssGSEA method to investigate tumor infiltration through the analysis of 24 different immune cell types. The Spearman correlation algorithm was employed to compare immune cell infiltration levels between subgroups with high and low TFRC gene expression and to evaluate the strength of association between TFRC gene expression and the concentrations of the 24 distinct immune cell types. The relationship between TFRC gene expression and immune infiltration, as well as the association between infiltrating levels of immune cells and the values obtained in different subgroups of TFRC gene expression, were analyzed using the module provided by the "Xiantao tool" based on the findings associated with immune infiltration.

## Patients and tissue samples

The CC tissues and the adjacent paracancerous tissues were obtained from the Department of Pathology, Second Affiliated Hospital, School of Medicine, Zhejiang University. The present study adhered to the ethical principles outlined in The Declaration of Helsinki. Study procedures were approved by the Ethics Committee of the Second Affiliated Hospital of Zhejiang University School of Medicine, Hangzhou, China (approval no: 2023-1138). The Clinical Research Ethics Committee of the Second Affiliated Hospital, Zhejiang University School of Medicine, provided ethical approval and waived informed consent. A total of 33 patients with CC who had undergone surgical resection at the Second Affiliated Hospital of Zhejiang University School of Medicine were included in the present study. CC samples and the corresponding medical information were collected from all patients.

## Immunohistochemistry staining

The protocol for IHC was essentially as we described previously (*Li et al., 2022*).
**Table 1 Abbreviations and their respective full forms.**

| Acronyms | Full name |
|---|---|
| CC | Cervical cancer |
| TFRC | Transferrin receptor protein 1 |
| TCGA | The Cancer Genome Atlas |
| GTEx | Genotype Tissue Expression |
| LASSO | Least absolute shrinkage and selection operator |
| FRGs | Ferroptosis-related genes |
| GSEA | Gene set enrichment analysis |
| ssGSEA | Single-sample GSEA |
| OS | Overall survival |
| GEO | Gene Expression Omnibus |
| DEGs | Differentially expressed genes |
| WGCNA | Weighted gene co-expression network analysis |
| GO | Gene Ontology |
| KEGG | Kyoto Encyclopedia of Genes and Genomes |
| KM | Kaplan–Meier |
| HPA | Human Protein Atlas |
| DSS | Disease Specific Survival |
| PFI | Progress Free Interval |
| HRP | Horseradish peroxidase |
| cDNA | Complementary DNA |

## Quantitative real-time PCR

Total RNA was isolated utilizing the TRIzol reagent manufactured by Invitrogen (United States). The synthesis of complementary DNA (cDNA) was performed using the PrimeScript RT kit provided by Takara. Quantification of mRNA levels was performed using qRT-PCR. The primers were designed by Shanghai Bio-Tech, as follows:

TFRC: forward, 5′-AGGTCAAAGACAGCGCTCAA-3′ and reverse, 5′-GCCACATAACCCCCAGGATT-3′.

Actin: forward, 5′-AGCCTTCCTTCCTGGGCAT-3′ and reverse, 5′-CTGTGTTGGCGTACAGGTCT-3′.

## Statistical analysis

Statistical and bioinformatics analyses were conducted using R software (version 4.2.0). We performed the Wilcoxon rank-sum test to examine the differential gene expression of TFRC in CC and normal tissues. Additionally, the Kruskal-Wallis test, logistic regression analysis, and Wilcoxon rank-sum test were used to investigate the relationship between TFRC gene expression and clinicopathological characteristics. The statistical significance of the observed variations was assessed through appropriate methods including the unpaired Student's t-test, Spearman's correlation analysis, Chi-square test, and Yates'

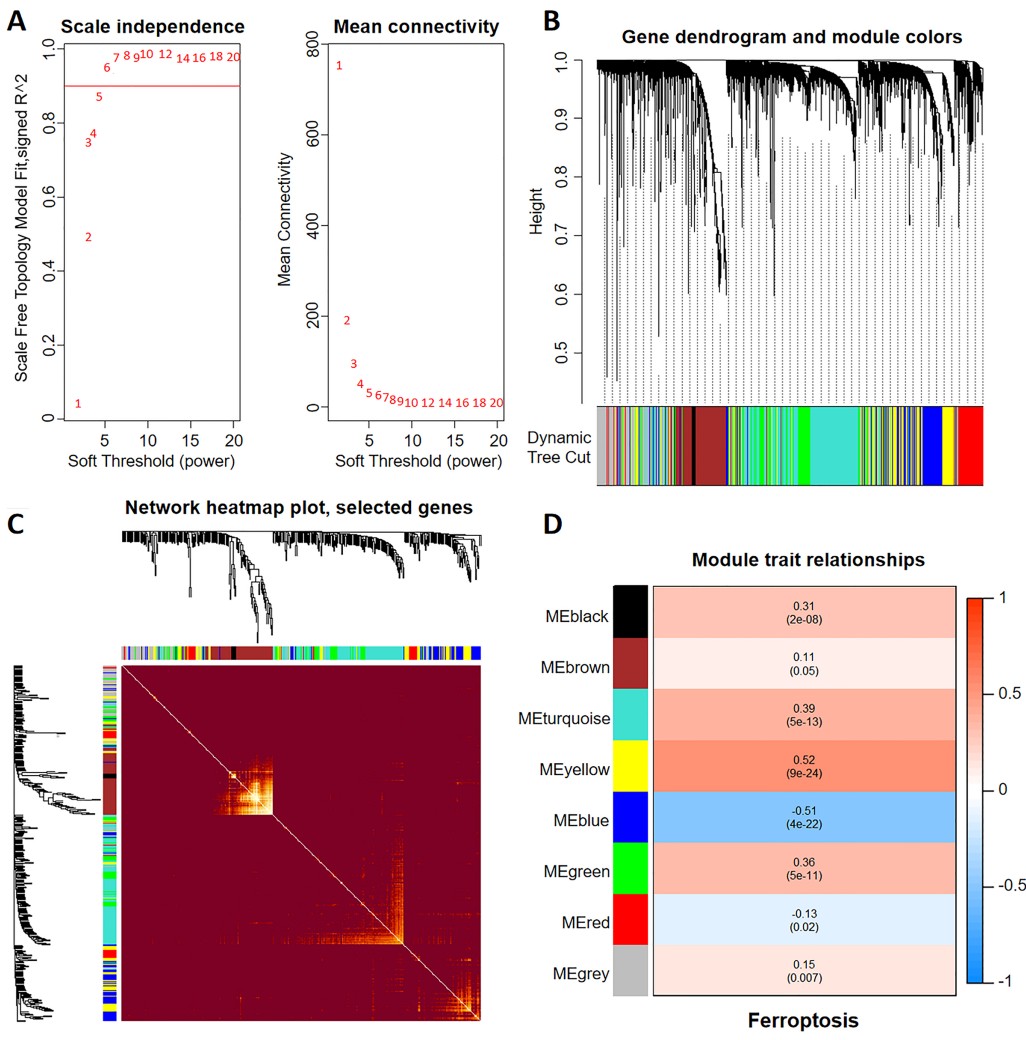

**Figure 1  Discovery of ferroptosis-related genes using WGCNA.** (A) The patterns and variations in the fit of the scale-free topology model and average connectivity are accompanied by soft threshold distribution and trends. (B) Gene clustering across various modules employing dynamic tree cut and merged dynamic approaches. The gray modules represent genes that remained unclassified. (C) Visualization of the topological overlap matrix of the genes using a heatmap. (D) Correlation between multiple modules and ferroptosis. Red signifies a positive correlation, while blue indicates a negative correlation. The numbers in parentheses represent the *P* values.   

correction, depending on the nature of comparisons conducted. Finally, Table 1 presents the complete list of abbreviations in English.

# RESULTS

## Identification of ferroptosis-related modules in CC using WGCNA

To construct a WGCNA network, we initially calculated β and estimated the co-expression similarity to compute the adjacency. Network topology analysis was performed using the WGCNA PickSoft threshold function. Subsequently, β was set to 6 as the scale independence reached 0.9 and exhibited relatively high average connectivity (Fig. 1A). The gene network and distinct modules were established using the one-step network

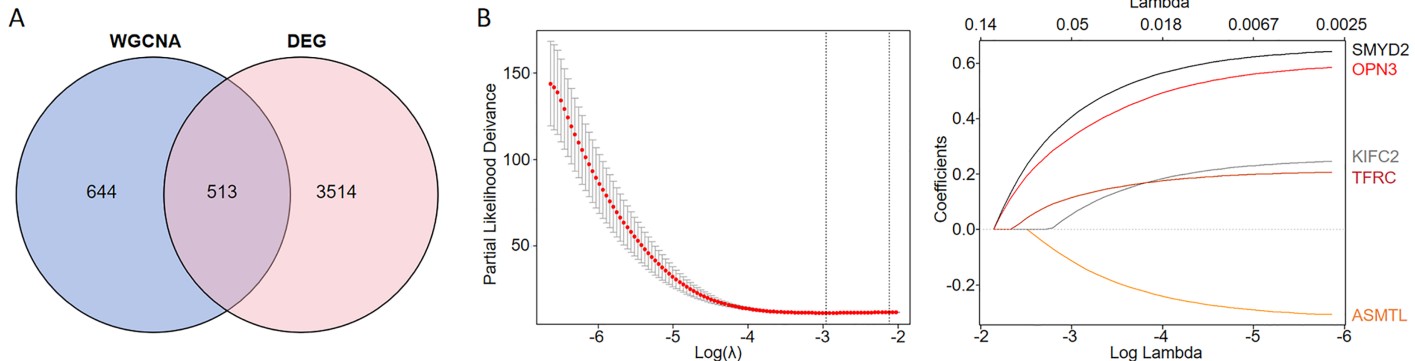

**Figure 2** **Identification of CC prognostic genes associated with ferroptosis using LASSO Cox regression.** (A) Venn diagram illustrating the gene counts obtained from overlapping the differentially expressed genes and those identified in the WGCNA analysis. (B) LASSO regression analysis of genes associated with ferroptosis.

construction function in the 'WGCNA' R package. Specifically, a WGCNA network was constructed, resulting in the generation of eight gene co-expression modules (Fig. 1B). The heatmap in Fig. 1C shows the topological overlap matrix of all the analyzed genes. There was a substantial degree of independence between these modules, implying relative independence in gene expression. Furthermore, the yellow modulus significantly correlated with ferroptosis (Fig. 1D).

## Identification of prognostic genes using LASSO analysis

We conducted an overlap analysis between the identified DEGs and the genes corresponding to the ferroptosis-related modules generated. A total of 3,514 DEGs and 644 ferroptosis-related module genes were selected for further analysis (Fig. 2A). Subsequently, using LASSO Cox regression analysis, we identified five key genes with the best prognostic value–KIFC2, TFRC, SMYD2, OPN3, and ASMTL. This method helped in reducing the dimensionality and calculating the correlation coefficients among genes (Fig. 2B).

## Functional enrichment analysis of the ferroptosis-related genes identified in the multigene signature

To investigate the biological functions of DEGs related to ferroptosis, we conducted GO annotation and KEGG enrichment analyses. GO analysis revealed significant enrichment in several biological processes, including chromosome segregation, organelle fission, nuclear division, nuclear chromosome segregation, and sister chromatid segregation. Among the cellular components, the enriched terms mainly included spindle, chromosomal region, and condensed chromosomal structures. Among the molecular functions, there was significant enrichment of the DNA-binding transcription activator activity, tubulin binding, microtubule binding, and cyclin-dependent protein serine kinase regulator activity (Figs. 3A, 3C and Tables 2, S1). Additionally, the GO analysis suggested that changes in the chromatin endoreduplication and segmentation complex were enriched in nuclear division, chromatin binding, and condensed chromosomes. Furthermore, KEGG enrichment analysis indicated the involvement of seven main

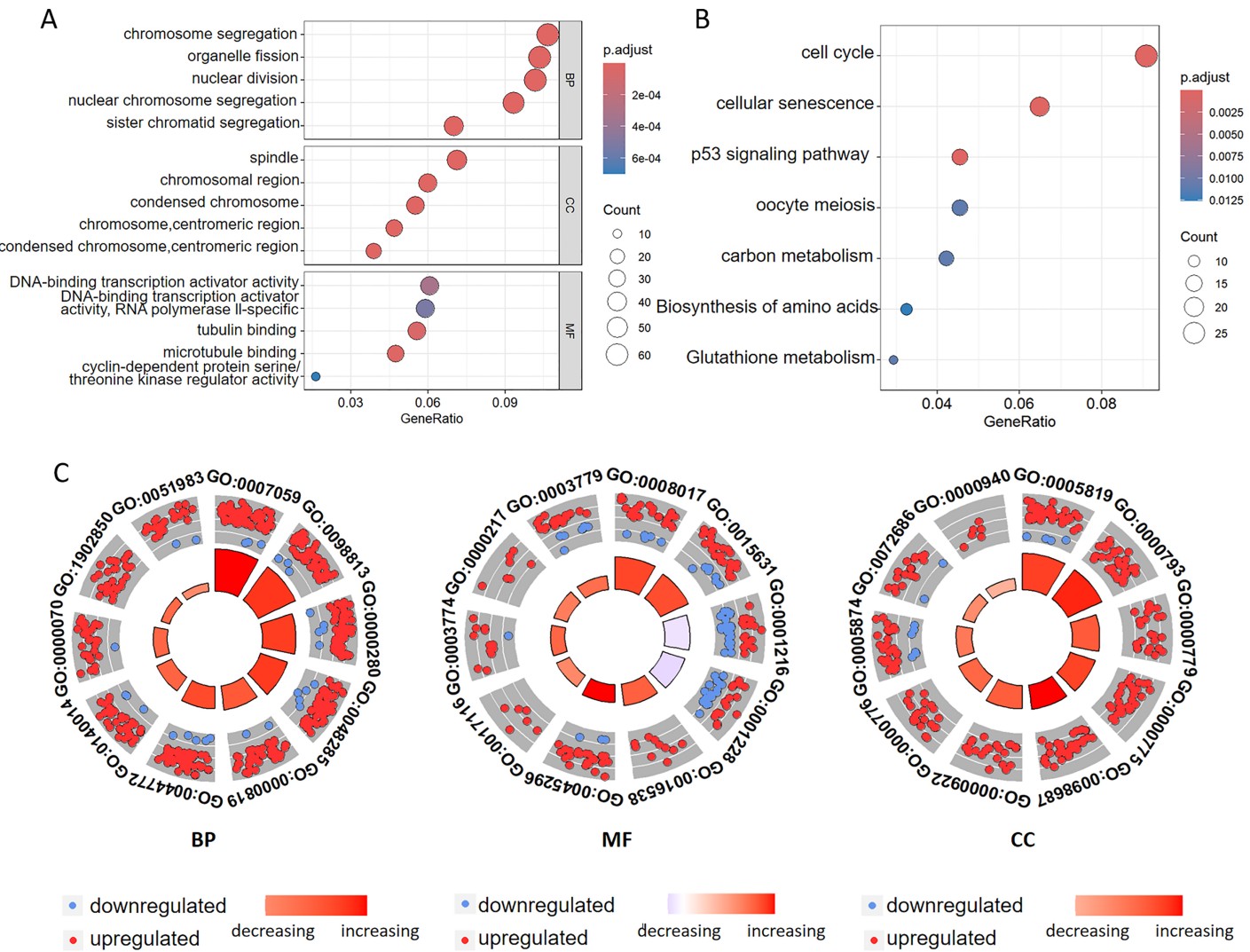

**Figure 3 Functional annotation of the ferroptosis-related genes identified in the multigene signature.** (A) GO and (B) KEGG pathway enrichment analyses of the ferroptosis-related genes identified in the multigene signature. (C) The top ten terms derived from the GO enrichment analysis are presented. BP represents biological processes, MF denotes molecular functions, and CC indicates cellular components. Red and blue dots symbolize upregulated and downregulated genes, respectively. The size and color of the sectors correspond to the adjusted $p$-value (adj. $P$. val) and standard score (z-score) of each respective GO term.

**Table 2 Supplementary information of GO analysis.**

| Category | ID | Term | adj. $P$. val |
|---|---|---|---|
| BP | GO:0007059 | Chromosome segregation | 4.584239e−21 |
| BP | GO:0098813 | Nuclear chromosome segregation | 6.756948e−21 |
| BP | GO:0000280 | Nuclear division | 1.559538e−18 |
| BP | GO:0048285 | Organelle fission | 4.084407e−17 |
| BP | GO:0000819 | Sister chromatid segregation | 9.041266e−16 |
| BP | GO:0044772 | Mitotic cell cycle phase transition | 2.740270e−15 |
| BP | GO:0140014 | Mitotic nuclear division | 1.011946e−13 |

(Continued)

| Category | ID | Term | adj. *P*. val |
|---|---|---|---|
| BP | GO:0000070 | Mitotic sister chromatid segregation | 7.210457e−13 |
| BP | GO:1902850 | Microtubule cytoskeleton organization involved in mitosis | 1.984252e−11 |
| BP | GO:0051983 | Regulation of chromosome segregation | 3.476946e−11 |
| MF | GO:0008017 | Microtubule binding | 1.820936e−05 |
| MF | GO:0015631 | Tubulin binding | 4.410044e−05 |
| MF | GO:0001216 | DNA-binding transcription activator activity | 3.252502e−04 |
| MF | GO:0001228 | DNA-binding transcription activator activity,RNA polymerase ll-specific | 5.240980e−04 |
| MF | GO:0016538 | Cyclin-dependent protein serine/threonine kinase regulator activity | 7.018462e−04 |
| MF | GO:0045296 | Cadherin binding | 2.429292e−03 |
| MF | GO:0017116 | Single-stranded DNA helicase activity | 8.706579e−03 |
| MF | GO:0003774 | Cytoskeletal motor activity | 9.674017e−03 |
| MF | GO:0000217 | DNA secondary structure binding | 9.674017e−03 |
| MF | GO:0003779 | Actin binding | 1.820936e−05 |
| CC | GO:0005819 | Spindle | 1.561395e−09 |
| CC | GO:0000793 | Condensed chromosome | 1.561395e−09 |
| CC | GO:0000779 | Condensed chromosome, centromeric region | 1.302102e−07 |
| CC | GO:0000775 | Chromosome, centromeric region | 1.794554e−07 |
| CC | GO:0098687 | Chromosomal region | 2.713408e−07 |
| CC | GO:0000922 | Spindle pole | 3.806959e−07 |
| CC | GO:0000776 | Kinetochore | 2.100519e−06 |
| CC | GO:0005874 | Microtubule | 1.329826e−05 |
| CC | GO:0072686 | Mitotic spindle | 5.893815e−05 |
| CC | GO:0000940 | Outer kinetochore | 6.416539e−05 |

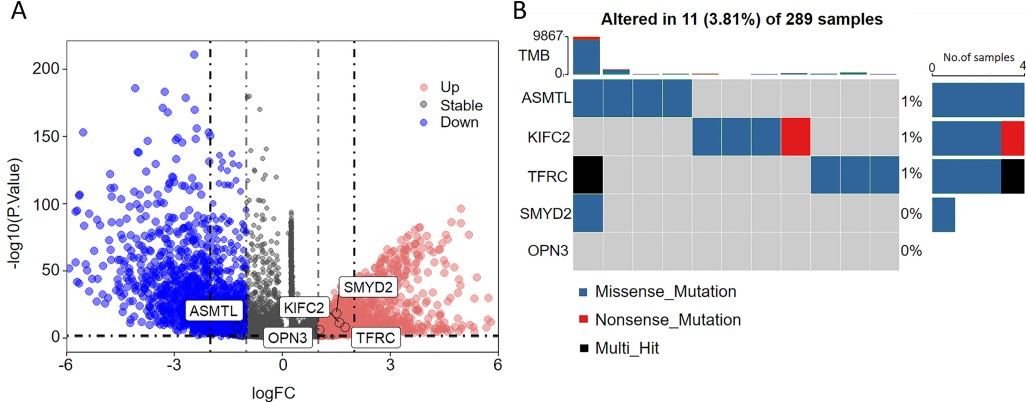

**Figure 4 Assessment of the involvement of the identified pivotal ferroptosis-related genes in cervical cancer.** (A) Volcano plot depicting the genes that were significantly upregulated (indicated in red) and downregulated (indicated in green) in the tumor group, as compared to that in the normal group. Log$_2$ (fold-change) > |1| and $P < 0.05$ were used as cut-off criteria to determine statistical significance. (B) Mutational profile of the pivotal molecules in 289 cervical cancer tissue samples.

pathways: cell cycle, cellular senescence, the p53 signaling pathway, oocyte meiosis, carbon metabolism, amino acid biosynthesis, and glutathione metabolism (Fig. 3B and Table S2). These results align with the established functions associated with ferroptosis in CC, thereby corroborating our findings.

## FRGs in the multigene signature displayed an association with prognostic value for CC

As seen in the volcano map given in Fig. 4A, the tumor samples displayed significant upregulation of KIFC2, TFRC, SMYD2, and OPN3 compared to the normal samples. Conversely, ASMTL was significantly downregulated in the tumor samples when compared to the normal samples. Additionally, we investigated the mutation landscape of the FRGs involved in CC. Among the 289 samples analyzed, 3.81% exhibited mutations in at least one key molecule. ASMTL, KIFC2, and TFRC displayed low mutation frequencies, whereas SMYD2 and OPN3 showed no mutations in the CC samples. The waterfall plot shown in Fig. 4B, which represents the mutation landscape of the five signature molecules, revealed that missense mutations were predominant. Furthermore, patients with CC who displayed low expression levels of KIFC2, TFRC, SMYD2, and OPN3 had a significant survival advantage over those with high expression levels of the same (Fig. 5A). Interestingly, high ASMTL expression conferred a significant survival advantage. Moreover, the expression levels of KIFC2, TFRC, SMYD2, and OPN3 were significantly higher in the CC tissues than in the normal tissues (Fig. 5B). Immunohistochemical staining results for the five genes obtained from the Human Protein Atlas database revealed that the tumor group displayed notably higher protein expression levels of KIFC2, TFRC, SMYD2, and OPN3 than the normal group (http://www.proteinatlas.org/) (Fig. 5C); conversely, ASMTL was downregulated in the tumor group when compared to that in the normal group. The mRNA expression levels of the five genes in the external validation datasets (GSE63514 and GSE63678) demonstrated similar findings to those in TCGA. With the exception of ASMTL, we observed upregulated mRNA levels of KIFC2, SMYD2, TFRC, and OPN3 in tumor tissues within these datasets (Figs. 5D and 5E).

## The FRGs-based prognostic model for CC

The patients with CC were stratified into high-risk and low-risk groups. Kaplan-Meier survival analysis demonstrated that the high-risk group exhibited poorer prognostic outcomes compared to the low-risk group within the TCGA dataset (Fig. 6A). Consistent with the findings from the TCGA training set, the high-risk groups demonstrated poor prognostic outcomes in the validation sets. The patients were stratified into high-risk and low-risk cohorts using a carefully determined cutoff value for the risk score, which was efficiently established through the MaxStat R package (Fig. 6B). Additionally, we examined the distribution of expression patterns of five FRGs, risk scores, survival time, and survival status in the TCGA dataset. Our results demonstrated the significant prognostic value of the ferroptosis-related signature (Fig. 6C).

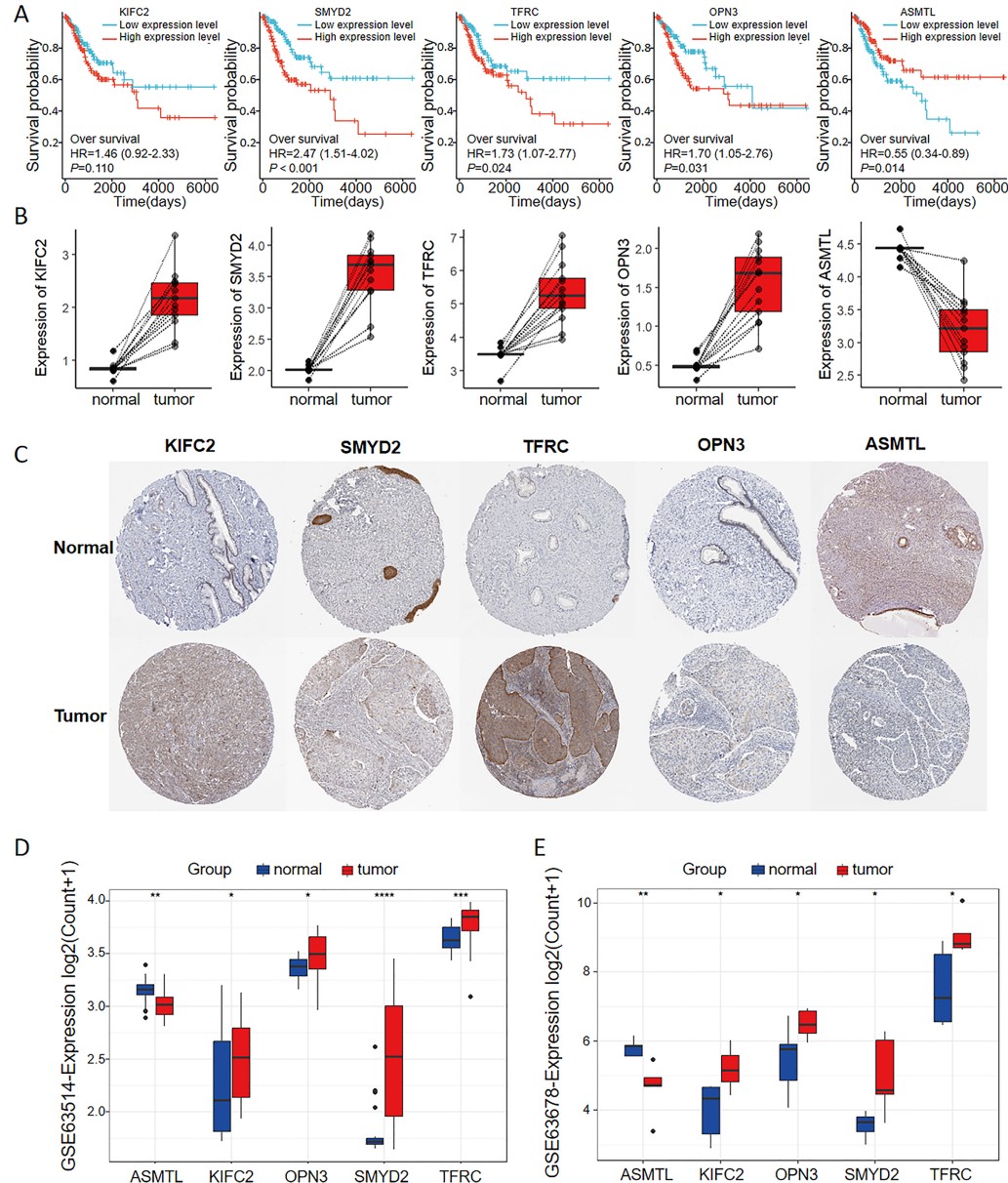

**Figure 5 Multi-omics analyses of the genes in the ferroptosis-related multigene signature.** (A) Elevated expression of KIFC2, TFRC, SMYD2, and OPN3, coupled with diminished expression of ASMTL in CC tissues, prognosticated unfavorable outcomes. (B) The mRNA and protein expression levels of KIFC2, TFRC, SMYD2, and OPN3 were significantly elevated in CC tissues compared to normal tissues in the TCGA and HPA databases. In contrast, ASMTL exhibited decreased expression. (C) Image credit: Human Protein Atlas. Normal KIFC2 available from version 23.0. https://www. proteinatlas.org/ENSG00000167702-KIFC2/tissue/cervix. Tumor KIFC2 available from version 23.0. https://www.proteinatlas.org/ENSG00000167702-KIFC2/pathology/cervical+cancer. Normal SMYD2 available from version 23.0. https://www.proteinatlas.org/ENSG00000143499-SMYD2/tissue/cervix. Tumor SMYD2 available from version 23.0. https://www.proteinatlas.org/ENSG00000143499-SMYD2/pathology/cervical+cancer. Normal TFRC available from version 23.0. https://www.proteinatlas.org/ENSG00000072274-TFRC/tissue/cervix. Tumor TFRC available from version 23.0. https://www.proteinatlas.org/ENSG00000072274-TFRC/pathology/cervical+cancer. Normal OPN3 available from version 23.0. https://www.proteinatlas.org/ENSG00000054277-OPN3/tissue/cervix. Tumor OPN3 available from version 23.0. https://www.proteinatlas.org/ENSG00000054277-OPN3/pathology/cervical+cancer. Normal ASMTL available from version 23.0. https://www.proteinatlas.org/ENSG00000169093-ASMTL/tissue/cervix. Tumor ASMTL available from version 23.0. https://www.proteinatlas.org/ENSG00000169093-ASMTL/pathology/cervical+cancer. License: CC BY SA 3.0.

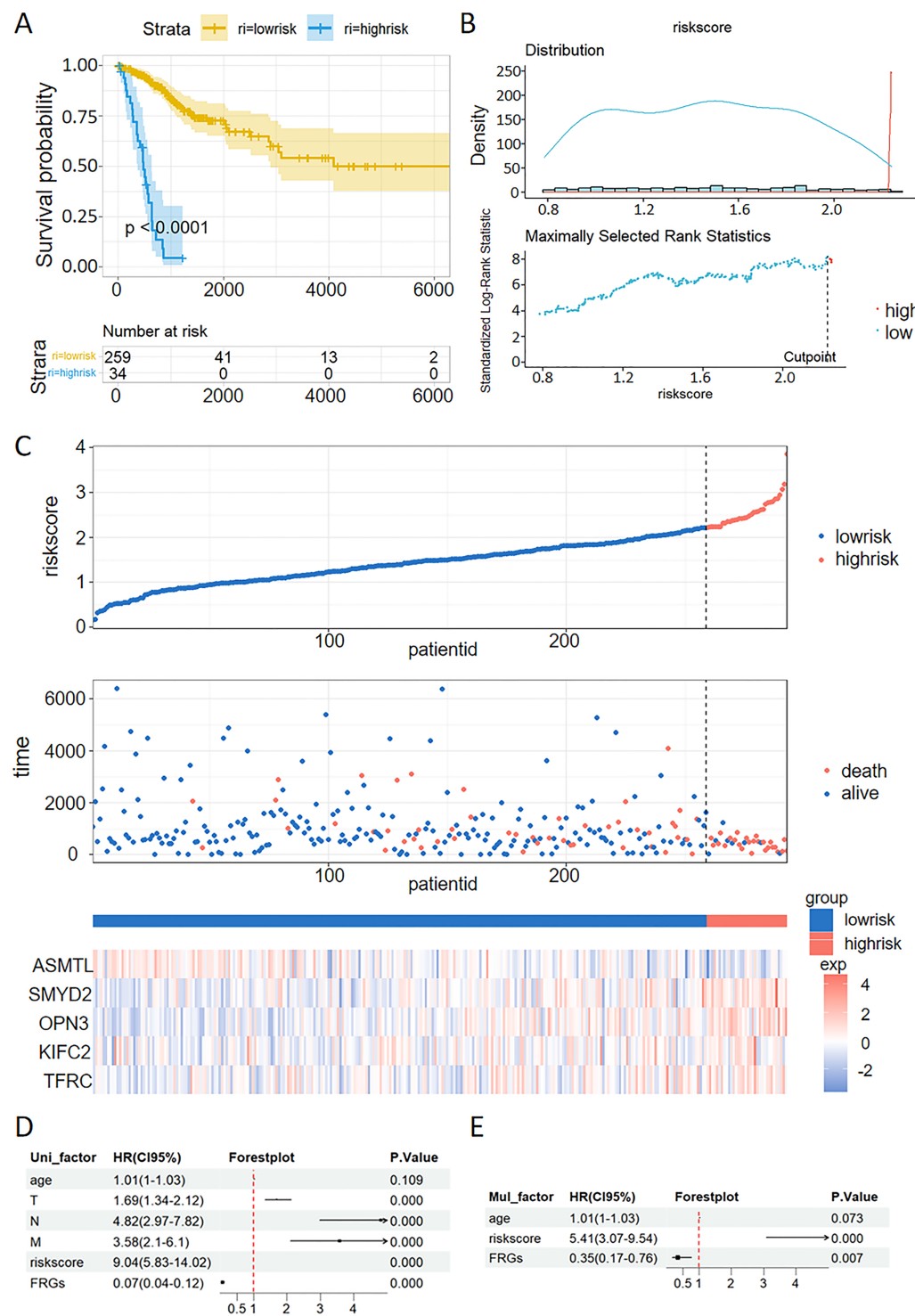

**Figure 6 Validation of a ferroptosis-related gene prognostic model and the predictive ability of a risk score and clinical factors for prognosis in patients with CC.** (A) Survival analysis of high- and low-risk groups in the TCGA datasets. (B) Optimal cutoff points for categorizing risk scores into low and high groups. (C) Distribution of expression profiles of 5 hub genes, the risk score, and survival status in TCGA dataset. (D and E) Univariate (D) and multivariate (E) Cox regression analyses were used to analyze correlations between OS and other clinical variables.

## Univariate and multivariate Cox analysis of the VMTRG signature

Univariate and multivariate Cox regression analyses were performed to establish if the five-gene signature is an independent predictor for OS of CC patients. Univariate Cox regression analysis of TCGA data revealed that T, N, M, riskscore and FRGs are independent prognostic variables for CC patients (Fig. 6D). Multivariate Cox regression analysis also showed that riskscore and FRGs are independent prognostic factors for CC patients in the TCGA dataset (Fig. 6E).

## Correlation between TFRC gene expression and whole gene expression patterns

We conducted an analysis of the TFRC gene expression profile to enhance our comprehension of the biological significance of the TFRC gene in CC. The gene expression heat map presented the top nine genes exhibiting aberrant expression levels (with a logFC > |1| and $P < 0.05$) (Fig. 7A). To investigate the biological and functional pathways associated with differential TFRC gene expression, we employed GSEA using data from TCGA. This analysis allowed us to determine the pathways that distinguish between high- and low-TFRC gene expression groups. The enrichment signaling pathway that exhibited the highest relevance to TFRC gene expression was selected based on the calculated normalized enrichment scores. The GSEA analysis revealed a predominant concentration of the TFRC gene expression phenotype in the ferroptosis, diseases of immune system, jak stat signaling pathway, ECM affiliated, RHO GTPases activate NADPH oxidases, JNK pathway, and condensation of chromosomes (Fig. 7B). Our findings indicate that TFRC likely plays a significant role in the progression of CC.

## Prognostic relevance of TFRC expression in CC

The TNM stage has long been recognized as an independent prognostic factor for CC survival. The T refers to the primary tumor, N refers to regional lymph node involvement, and M refers to distant metastasis. The term "OS event" denotes the duration spanning from treatment commencement to the patient's demise. The relationship between TFRC expression and clinical parameters was assessed using the Kruskal-Wallis and Wilcoxon signed-rank tests. Elevated levels of TFRC expression were found to be associated with higher T stage, and N stage, as well as occurrence of OS events (Figs. 8A–8C). Table 3 displays the correlation between the clinicopathological characteristics of 306 CC patients and their TFRC protein levels. Patients exhibiting high TFRC expression demonstrated poorer survival outcomes. These findings provide evidence suggesting an association between increased TFRC expression and advanced tumor T stage, thus indicating a crucial role of TFRC in CC prognosis. We developed a clinical prognostic risk score for CC incorporating T stage, N stage, M stage, clinical stage, histological grade, age, and TFRC expression (Fig. 8D). Additionally, a calibration chart was utilized to evaluate the accuracy of the model's predictions (Fig. 8E). TFRC expression has the potential to offer improved accuracy in predicting patients' survival probabilities for 3- and 5-year intervals. In general, there was a demonstrated correlation between TFRC expression and the prognosis of patients with CC.

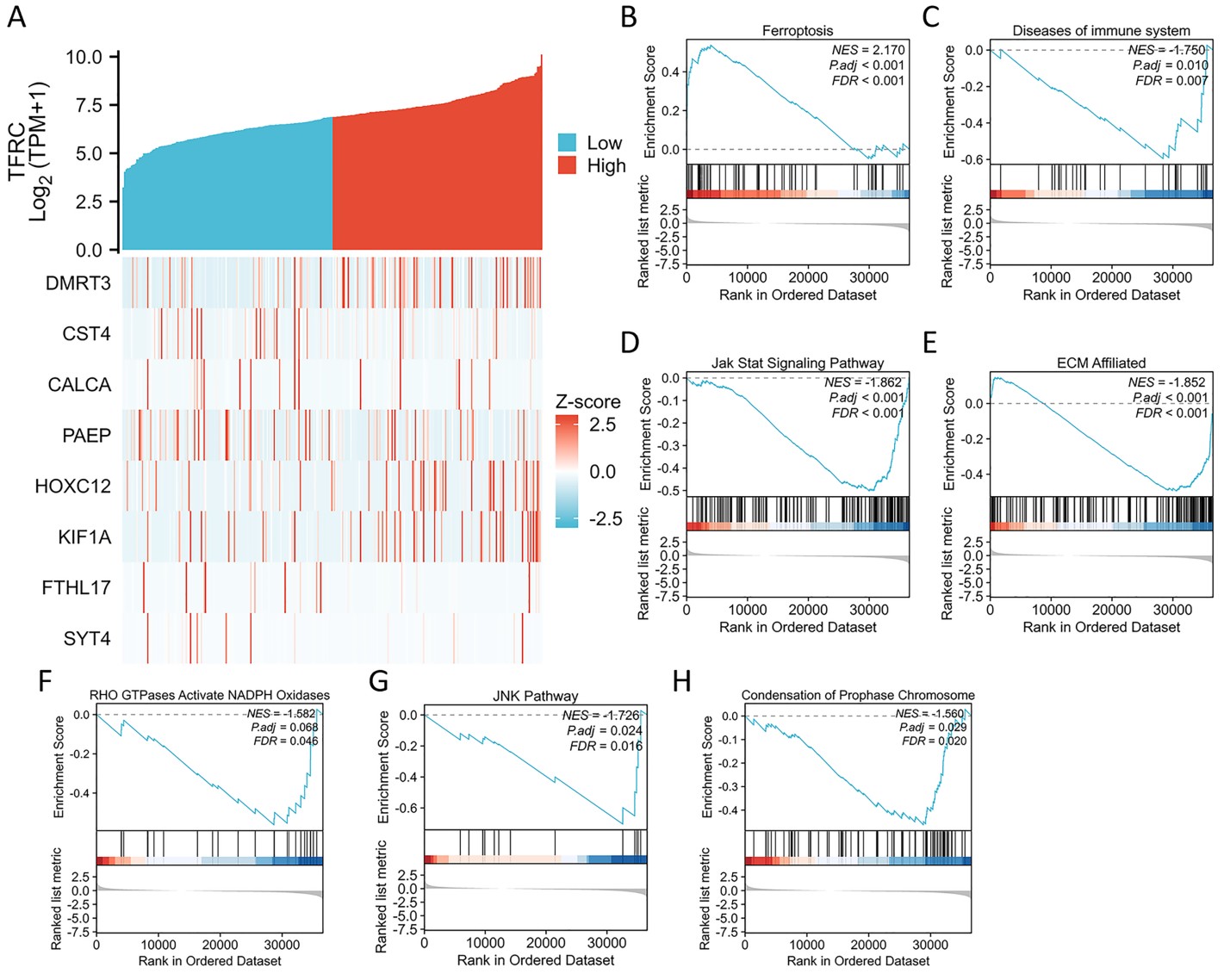

**Figure 7 Differential expression of the TFRC gene and GSEA.** (A) The expression level of the TFRC gene was utilized to create a heat map displaying nine genes that exhibited upregulation or downregulation. (B) GSEA results indicated that CC with high TFRC mRNA level was significantly enriched with ferroptosis. (C–H) GSEA results indicated that CC with low TFRC mRNA level was significantly enriched with diseases of immune system, Jak stat signaling pathway, ECM affiliated, RHO GTPases activate NADPH oxidases, JNK pathway, and condensation of prophase chromosome.

Overall, TFRC expression was shown to correlate with the prognosis of patients with CC. The figures illustrate the associations between TFRC expression and prognosis indicators using data from the TCGA database, including DSS and PFI. Increased TFRC expression correlated with poor outcomes in terms of DSS (HR = 2.10 (1.20–3.66), $P = 0.009$, Fig. 8F) and PFI (HR = 1.93 (1.19–3.13), $P = 0.007$, Fig. 8G).

## The association between TFRC gene expression and immune cell infiltration

This study investigated and analyzed the relationship between TFRC gene expression and 24 distinct immune cell subtypes in CC. There was a significant positive correlation

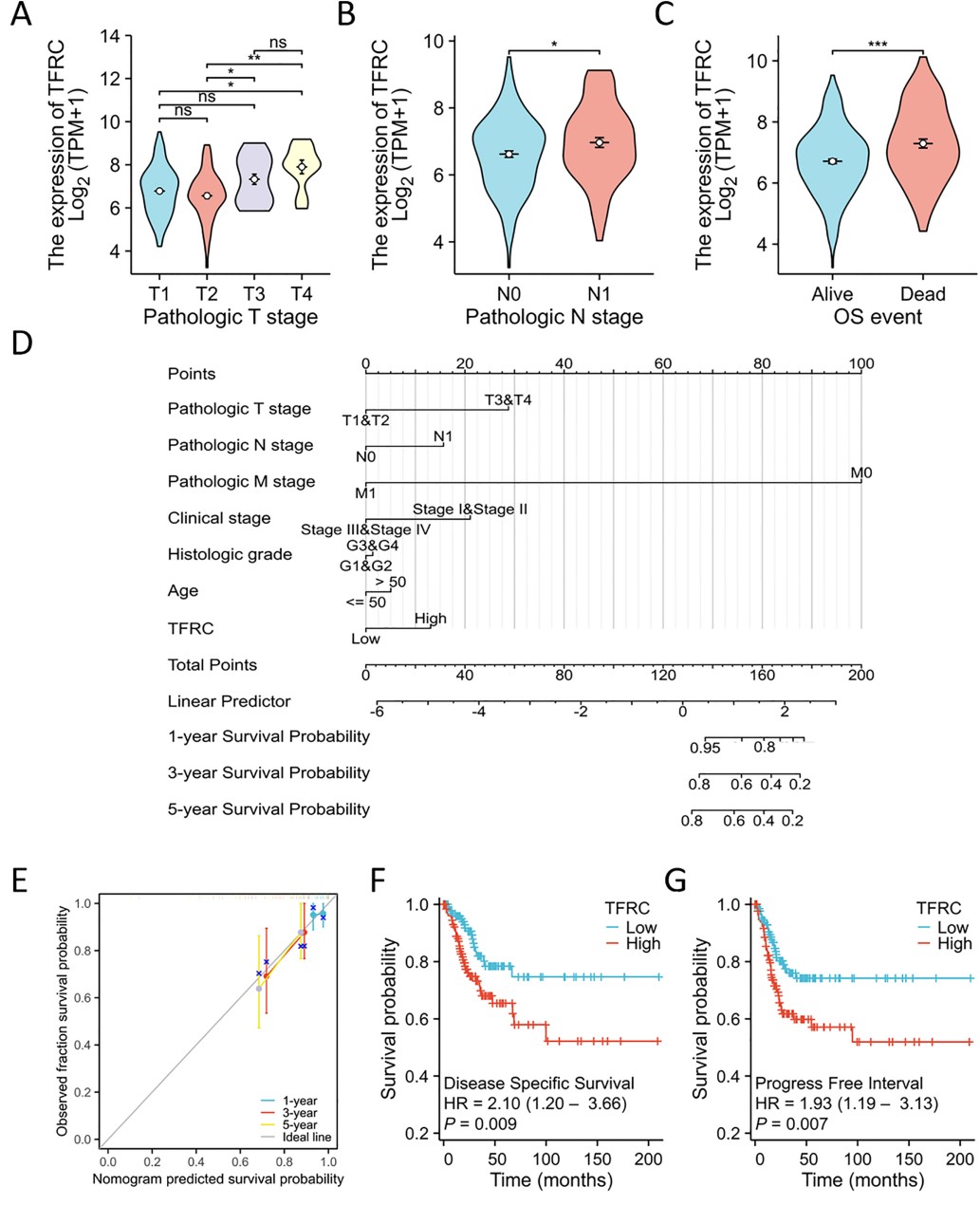

**Figure 8 TFRC expression prognostic analysis.** (A–C) The relationship between TFRC expression and T stage, N stage, OS event. (D) A multivariate analysis nomogram utilizing clinical features associated with the expression of TFRC. (E) The calibration chart illustrates the predictive accuracy determined through multi-factor Cox regression analysis. (F and G) Patients exhibiting high TFRC expression displayed unfavorable prognosis indicators compared to those with low TFRC expression, PFI, DSS. $*p < 0.05$, $**p < 0.01$, and $***p < 0.001$, ns, no statistical difference.

between TFRC gene expression and infiltration of Th2 cells, Tgd cells, and Tcm cells, while a strong inverse correlation was observed with T cells, pDCs, and cytotoxic cells, among others (Fig. 9A). Subsequent analysis revealed significant variations in TFRC gene

**Table 3 Association of TFRC expression with clinicopathological characteristics in patients with CC.**

| Characteristics | Low expression of TFRC | High expression of TFRC | P value | Method |
|---|---|---|---|---|
| n | 153 | 153 | | |
| Pathologic T stage, n (%) | | | 0.016 | Yates' correction |
| T1 | 71 (29.2%) | 69 (28.4%) | | |
| T2 | 29 (11.9%) | 43 (17.7%) | | |
| T3 | 9 (3.7%) | 12 (4.9%) | | |
| T4 | 2 (0.8%) | 8 (3.3%) | | |
| Pathologic N stage, n (%) | | | 0.266 | Chisq test |
| N0 | 73 (37.4%) | 61 (31.3%) | | |
| N1 | 28 (14.4%) | 33 (16.9%) | | |
| Pathologic M stage, n (%) | | | 0.721 | Yates' correction |
| M0 | 65 (51.2%) | 51 (40.2%) | | |
| M1 | 5 (3.9%) | 6 (4.7%) | | |
| Clinical stage, n (%) | | | 0.231 | Chisq test |
| Stage I | 83 (27.8%) | 79 (26.4%) | | |
| Stage II | 39 (13%) | 30 (10%) | | |
| Stage III | 19 (6.4%) | 27 (9%) | | |
| Stage IV | 8 (2.7%) | 14 (4.7%) | | |
| OS event, n (%) | | | 0.021 | Chisq test |
| Alive | 125 (40.8%) | 109 (35.6%) | | |
| Dead | 28 (9.2%) | 44 (14.4%) | | |
| Age, n (%) | | | 0.100 | Chisq test |
| <= 50 | 101 (33%) | 87 (28.4%) | | |
| >50 | 52 (17%) | 66 (21.6%) | | |

expression levels across various infiltrating immune cell types, including neutrophils, NK CD56bright cells, pDCs, T cells, iDCs, DCs, cytotoxic cells, CD8 T cells, B cells, aDCs, Treg cells, and TH1 cells, among others (Figs. 9B–9D). Moreover, this study examined various functional subsets of T cells, including Tregs, exhausted T cells, Th1, Th2, Th9, Th17, Th22, and Tfh cells.

## Experimental verification of the expression of TFRC

The transferrin receptor TFRC serves as a crucial component of the channel responsible for facilitating the uptake of iron ions into cells. It plays a pivotal role in regulating cellular iron metabolism and maintaining iron balance. To investigate the variations in the mRNA and protein expression of TFRC between paracancerous tissue and CC tissues, we examined the expression levels of TFRC in tissues by using qRT-PCR and IHC (Figs. 10A–10C). All procedures for cDNA synthesis, and qPCR were designed to follow the MIQE guidelines. The findings indicate that TFRC showed significantly higher expression levels in CC tissues compared to the adjacent paracancerous tissues.

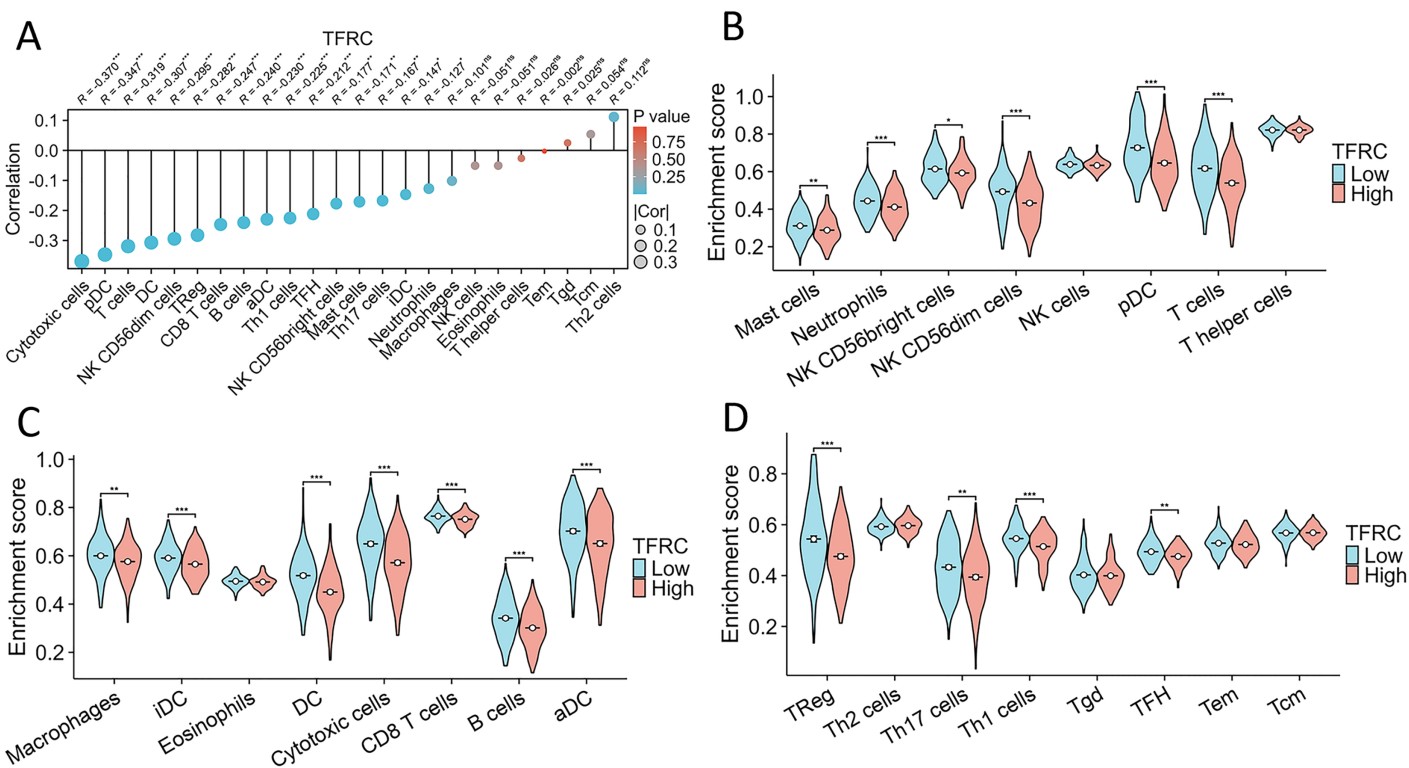

**Figure 9 Association between TFRC gene expression and infiltration of immune cells.** (A) The association between TFRC gene expression and the status of immune cell infiltration. (B–D) Variation in the extent to which specific immune cell subsets were enriched in the high- and low-expression groups of the TFRC gene. *$p < 0.05$, **$p < 0.01$, and ***$p < 0.001$.

## Prognostic significance of TFRC in CC

Patient follow-up commenced from the resection date and extended until October 2023, with survival or death serving as the definitive endpoint for overall survival assessment. The prognostic significance of TFRC was assessed using Kaplan-Meier (KM) survival analysis. Patients with CC who displayed low TFRC expression levels exhibited a significant survival benefit in comparison to those with high TFRC expression levels (Fig. 10D). This implies that TFRC holds promise as a prognostic biomarker in CC and could potentially be targeted for therapeutic interventions in CC treatment.

## DISCUSSION

Ferroptosis is a unique mode of cellular demise, distinct from conventional cell death pathways. Ferroptosis has demonstrated considerable potential in anticancer treatment. Iron dysregulation and oxidative stress are closely linked to the progression of CC. Hence, interventions aimed at inducing ferroptosis could present novel therapeutic strategies for managing CC. In this study, a novel prognostic signature comprising five FRGs, namely KIFC2, TFRC, SMYD2, OPN3, and ASMTL, was developed to predict the prognosis of CC patients. Unlike previous studies that exclusively relied on the LASSO algorithm and COX regression analysis, our research integrates validation using both internal and external data to identify potential specificity in the development of prognostic markers. Importantly, we

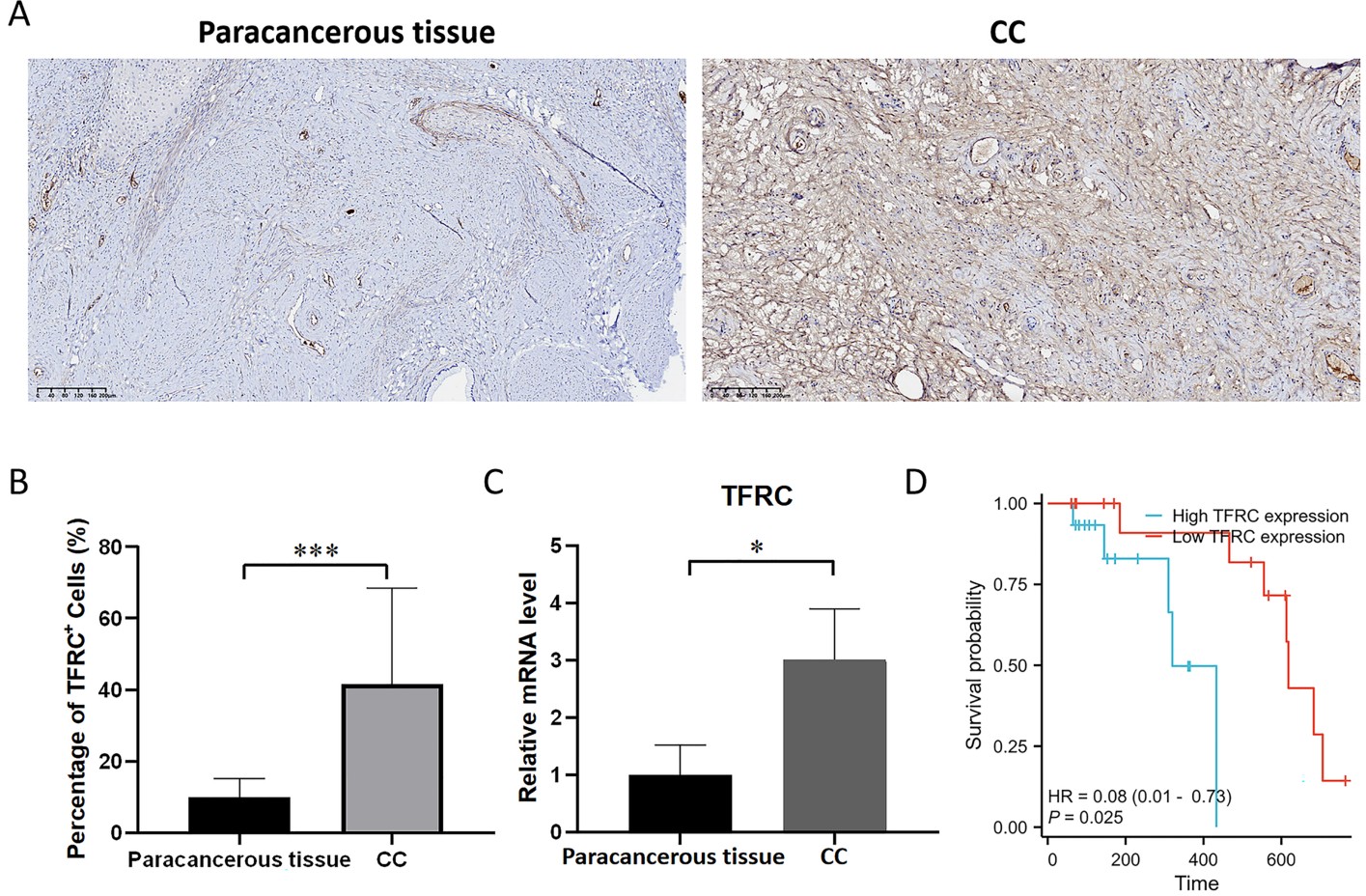

**Figure 10 The mRNA and protein expression levels of TFRC.** (A) The protein expression level of TFRC was examined by IHC. (B) Relative TFRC optical density. (C) The mRNA expression level of TFRC was examined by qRT-PCR. (D) The KM plot illustrates the survival outcomes of CC patients categorized by high (blue line) or low (red line) TFRC expression levels. 

investigated the mRNA and protein level of TFRC through qRT-PCR and IHC. Additionally, we explored the association between TFRC and the prognostic outcomes in these patients, which carries substantial implications for our future research and conclusions.

KIFC2 is crucial for tumor development and drug resistance and has been identified as a potential biomarker or therapeutic target for cancer treatment (*Liu et al., 2023*). However, there is a need for further investigation to elucidate the mechanisms by which KIFC2 affects ferroptosis in various tumor types. SMYD2 can methylate lysine residues in both histone and non-histone proteins associated with cancer, thereby playing a crucial role in tumorigenesis (*Yadav & Singh, 2023*). SMYD2 expression is upregulated in CC cells, suggesting its involvement in tumor metabolism. OPN3 has been implicated in tumor metastasis and drug sensitivity. According to *Xu et al. (2020a)*, OPN3 enhances tumor metastasis in lung adenocarcinoma. *Sui et al. (2021)* discovered that ASMTL-AS1 positively regulates SAT1, promoting ferroptosis and stabilizing SAT1 mRNA *via* the recruitment of U2AF2. These findings shed light on a novel molecular mechanism

involved in the progression of CC. In the present study, ASMTL was identified as a protective factor with high expression levels in CC. It is currently uncertain whether these four genes impact the prognosis of cervical cancer patients by modulating iron metabolism, given the limited research on these genes.

TFRC encodes a classical transferrin receptor that is essential for cellular iron uptake. TFRC has been reported to be significantly upregulated in a variety of tumors, including bladder cancer and hepatocarcinoma, where it plays a crucial role in promoting tumor cell proliferation, invasion, metastasis, as well as conferring resistance to radiotherapy and chemotherapy (*Tang et al., 2024*; *Wang et al., 2023a*). The study successfully validated the correlation between TFRC protein levels and clinical pathological features in a cohort of CC patients. Analysis of clinical data demonstrated a notable correlation between TFRC expression levels and the T stage of cervical cancer patients. Progressive elevation of CC expression in advanced-stage T-phase patients is observed. Patients with CC who have low TFRC expression demonstrate prolonged survival compared to those with high expression levels. These findings are consistent with previous studies on bladder cancer, oral squamous cell carcinoma, and lung squamous cell carcinoma, collectively indicating the involvement of TFRC as an oncogene in malignant tumor progression, resulting in a worsened prognosis (*Arora et al., 2023*; *Miao et al., 2022*; *Tang et al., 2024*). Therefore, TFRC can function as an independent prognostic indicator for overall survival (OS). Thus, our study offers valuable insights and practical significance by suggesting TFRC as a promising molecular biomarker for CC, thereby enhancing the clinical management of future CC patients.

In order to enhance comprehension of TFRC function and its associated activation pathways, GSEA was conducted. The GSEA analysis unveiled TFRC involvement in regulating the malignant phenotype of CC, as well as its participation in pathways linked to immune deficiency, the JNK pathway, chromosome condensation, the RHOC GTPase cycle, and ECM receptor interactions associated with invasive functions. Prior research has shown that TFRC modulates mitochondrial fusion through regulation of the JNK pathway, which is consistent with our enrichment results (*Senyilmaz et al., 2015*). The enrichment of pathways related to ferroptosis in our functional analysis provided further validation for our results. Due to the vital role of ferroptosis in tumorigenesis, our comprehension of the mechanisms governing tumor advancement and prognosis in TFRC has notably progressed. There has been a growing interest in elucidating the factors influencing tumor susceptibility to ferroptosis. Nonetheless, the exact mechanism responsible for iron accumulation in tumors remains elusive and warrants investigation in forthcoming research. Particularly noteworthy is the significant impact of TFRC on immune-related pathways, indicating its function as an oncogene in shaping the immune landscape of tumors. Notably, TFRC extensively influences immune-related pathways, suggesting its role as an oncogene in modulating the immune microenvironment of tumors.

However, no previous studies have linked TFRC genes to immune cells in CC. Therefore, our study innovatively investigated and analyzed the association of TFRC expression in CC with 24 different immune cell subtypes. Our findings show that the TFRC gene expression level has a substantial and consistent relationship with immune cell

infiltration levels in CC. We also discovered that the expression of CD8+ T cell markers, including CD8, correlates negatively with TFRC expression. CD8+ T cells have the capacity to differentiate into cytotoxic T cells, enabling them to directly eliminate cancer cells. CD8+ T cells play a role in tumor invasion and progression in the tumor microenvironment. These findings indicate that TFRC plays a crucial role in the initiation and advancement of CC, as well as immunoregulatory processes, immune cell infiltration, and the efficacy of immunotherapy. Consequently, targeting TFRC presents a potential alternative strategy for tumor therapy.

Nonetheless, this study has certain limitations. Firstly, the lack of empirical data from public databases and potential contamination in tissues may have led to biased results. Secondly, limited access to clinical samples hindered the acquisition of sufficient clinical evidence to conclusively establish TFRC as an independent predictive factor in CC. Therefore, validation in future clinical trials is warranted.

# CONCLUSIONS

This study ingeniously combines macroscopic exploration with microscopic analysis to ascertain the screening direction of FRGs. Leveraging methodologies such as big data analysis, bioinformatics, and histopathology, it guarantees precise and efficient exploration of FRGs. The results of this study suggest that elevated TFRC expression in CC is linked to disease progression, an adverse prognosis, and dysregulated immune cell infiltration. We hope that the identified TFRC will help improve strategies for the personalized treatment of patients with CC and aid in improved treatment decisions.

## Funding
This work was supported by the National Natural Science Foundation of China (No. 81971291). The funders had no role in study design, data collection and analysis, decision to publish, or preparation of the manuscript.

## Grant Disclosures
The following grant information was disclosed by the authors:
National Natural Science Foundation of China: 81971291.

## Competing Interests
The authors declare that they have no competing interests.

## Author Contributions
- Xiujuan Shang analyzed the data, prepared figures and/or tables, authored or reviewed drafts of the article, and approved the final draft.
- Hongdong Wang analyzed the data, prepared figures and/or tables, authored or reviewed drafts of the article, and approved the final draft.
- Jin Gu performed the experiments, authored or reviewed drafts of the article, and approved the final draft.

- Xiaohui Zhao performed the experiments, authored or reviewed drafts of the article, and approved the final draft.
- Jing Zhang performed the experiments, authored or reviewed drafts of the article, and approved the final draft.
- Bohao Sun conceived and designed the experiments, authored or reviewed drafts of the article, and approved the final draft.
- Xinming Zhu conceived and designed the experiments, analyzed the data, authored or reviewed drafts of the article, and approved the final draft.

## Human Ethics

The following information was supplied relating to ethical approvals (*i.e.*, approving body and any reference numbers):

The Clinical Research Ethics Committee of The Second Affiliated Hospital, Zhejiang University School of Medicine approved the study to be performed within its facilities (Ethical Application Ref: 2023-1138).

## Data Availability

The raw measurements are available in the Supplemental File.

## Supplemental Information

Supplemental information for this article can be found online at http://dx.doi.org/10.7717/peerj.17842#supplemental-information.

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
