# Peer review of "Ferroptosis-related gene transferrin receptor protein 1 expression correlates with the prognosis and tumor immune microenvironment in cervical cancer"

_PeerJ, doi:10.7717/peerj.17842_

## Round 0.1 · original submission · Minor Revisions

Please address the concerns of both reviewers and amend the manuscript accordingly.

Reviewer 1 ·

Basic reporting

The manuscript by Shang et al. describes the role of TFRC, a ferroptosis-related gene in cervical cancer. Overall, the study is thorough and all aspects have been properly addressed. However, there are certain minor issues that need to be addressed and I have detailed them here.
1. The authors need to state the full form of TFRC gene, OS event, and qPCR in the Background, Methods, and Results section.
2. It is necessary to add more background about the FRGs. For example, the authors can include a couple of sentences FRGs, previous studies about how FRGs have been studied to target ferroptosis, and other details that authors deem fit.
3. The font size of the figure text and the axis labels need to be increased.
4. In Fig 1D. the color key on the right axis needs to describes which color represents the strongest and poorest correlation.
5. A reference needs to be added for the statement on line 249.
6. The meaning of BP, MF, and CC is unclear from Fig. 3 legend. Additionally, there are no black dots in the figure and the authors either need to change the word to blue or alter the color in the figure.
7. In Fig. 4, the color code of the normal group needs to be mentioned in the figure legend.
8. In Fig. 5, more description in the legend needs to be added depicting the inference obtained from the figures. In fig. 5A and in other figures, the word ‘expression level’ needs to be added to the ‘low’ and ‘high’ words.
9. The authors need to explain the terms T, N , M stage, and OS events in lines 309 and 315.
10. The manuscript contains numerous acronyms that does not have the full form or an explanation of the term associated with it. The issue needs to be resolved wherever applicable in the manuscript.
11. Fig. 7 does not have C panel although the legend consists of a description under C.
12. Fig. 9E-L does not have a legend and neither has it been explained in the manuscript. The authors should either remove it or add a description and its relevance.
13. The discussion should start with a brief background followed by a brief summary of the findings.
14. Some references have numbers while others have the names of the authors. It needs to be consistent throughout the manuscript.

Experimental design

The experiments are well-designed. I have one issue related to the results.

In the Results, the authors need to briefly describe the protocol they followed for synthesis of cDNA and qPCR reaction conditions.

Validity of the findings

All data has been provided and and conclusion are related to the research question.

Reviewer 2 ·

Basic reporting

- Introduction
The introduction provides a comprehensive overview of cervical cancer (CC), highlighting its prevalence, current challenges, and the potential role of ferroptosis in its treatment. The integration of recent studies and statistics helps establish the significance of the topic. However, some areas could be improved for clarity, conciseness, and depth.
line 51-52: "Nonetheless, these approaches have been ineffective in inducing tumor regression." - This sentence could benefit from elaboration on why these approaches are ineffective?
line 52: The phrase "persistent human papillomavirus infection" could be rephrased to "Persistent infection with human papillomavirus (HPV)" for clarity.
Fourth Paragraph: The transition to discussing ferroptosis-related treatment should be more explicit about why current therapies fail, thus necessitating new approaches.
Final Paragraph: Consider rephrasing to explicitly state the novelty of the study's approach and the significance of TFRC in the broader context of CC research.

Experimental design

The methods section is detailed and thorough, demonstrating a comprehensive approach to studying ferroptosis-related genes (FRGs) in cervical cancer (CC). The use of multiple datasets, robust statistical methods, and various bioinformatics tools enhances the study's validity. However, some areas could be improved for better clarity, reproducibility, and justification of the chosen methods.

Normalization of Microarray Data in line 94: "The microarray data were subjected to normalization..." - Specify the normalization method used (e.g., RMA, quantile normalization).

WGCNA Process in line 104: "We determined an appropriate soft thresholding power (β)..." - Provide the range of β values tested and the criteria for selecting the optimal value.

LASSO and SVM-RFE in line 113: "The LASSO logistic regression and Support Vector Machine-Recursive Feature Elimination methods..." - Explain the advantages of combining these methods and how they complement each other.

Functional Annotation in line 126: "GO enrichment analysis elucidated three aspects: biological processes, cellular components, and molecular functions." - Provide a table of significant GO terms and KEGG pathways identified in the supporting information material.

Validity of the findings

The conclusions are well stated, linked to original research question & limited to supporting results.

---

## Round 0.2 · accepted · Accept

All issues pointed by the reviewers were adequately addressed and revised manuscript is acceptable now.